# *p27*^Kip1^ Deficiency Impairs Brown Adipose Tissue Function Favouring Fat Accumulation in Mice

**DOI:** 10.3390/ijms24032664

**Published:** 2023-01-31

**Authors:** Ignacio Colon-Mesa, Neira Sainz, Patricia Corrales, María Collantes, Philipp Kaldis, José Alfredo Martinez, Gema Medina-Gómez, María Jesús Moreno-Aliaga, Xavier Escoté

**Affiliations:** 1Department of Nutrition, Food Science and Physiology and Center for Nutrition Research, School of Pharmacy and Nutrition, University of Navarra, Irunlarrea 1, 31008 Pamplona, Spain; 2Department of Basic Sciences of Health, Area of Biochemistry and Molecular Biology, Universidad Rey Juan Carlos, Alcorcon, 28933 Madrid, Spain; 3Nuclear Medicine Department, Clínica Universidad de Navarra, 31008 Pamplona, Spain; 4Department of Clinical Sciences, Lund University, Clinical Research Centre (CRC), P.O. Box 50332, SE-202 13 Malmö, Sweden; 5Lund University Diabetes Centre (LUDC), Lund University, SE-202 13 Malmö, Sweden; 6LAFEMEX Laboratory, Área de Bioquímica y Biología Molecular, Departamento de Ciencias Básicas de la Salud, Facultad de Ciencias de la Salud, Universidad Rey Juan Carlos, 28922 Alcorcón, Spain; 7IdISNA—Navarra Institute for Health Research, 31008 Pamplona, Spain; 8CIBER de Fisiopatología de la Obesidad y Nutrición (CIBEROBN), Instituto de Salud Carlos III, 28029 Madrid, Spain; 9Eurecat, Centre Tecnològic de Catalunya, Unitat de Nutrició i Salut, 43204 Reus, Spain

**Keywords:** brown adipose tissue, insulin resistance, obesity, *p27*

## Abstract

The aim of this work was to investigate the effect of the whole-body deletion of *p27* on the activity of brown adipose tissue and the susceptibility to develop obesity and glucose homeostasis disturbances in mice, especially when subjected to a high fat diet. *p27* knockout (*p27*^−/−^) and wild type (WT) mice were fed a normal chow diet or a high fat diet (HFD) for 10-weeks. Body weight and composition were assessed. Insulin and glucose tolerance tests and indirect calorimetry assays were performed. Histological analysis of interscapular BAT (iBAT) was carried out, and expression of key genes/proteins involved in BAT function were characterized by qPCR and Western blot. iBAT activity was estimated by ^18^F-fluorodeoxyglucose (^18^FDG) uptake with microPET. *p27*^−/−^ mice were more prone to develop obesity and insulin resistance, exhibiting increased size of all fat depots. *p27*^−/−^ mice displayed a higher respiratory exchange ratio. iBAT presented larger adipocytes in *p27*^−/−^ HFD mice, accompanied by downregulation of both *Glut1* and uncoupling protein 1 (UCP1) in parallel with defective insulin signalling. Moreover, *p27*^−/−^ HFD mice exhibited impaired response to cold exposure, characterized by a reduced iBAT ^18^FDG uptake and difficulty to maintain body temperature when exposed to cold compared to WT HFD mice, suggesting reduced thermogenic capacity. These data suggest that *p27* could play a role in BAT activation and in the susceptibility to develop obesity and insulin resistance.

## 1. Introduction

Over recent decades overweight/obesity has become a worldwide health concern [1], having rapidly increased in prevalence in the last 30 years [2]. Many factors are involved in the development of obesity, including genetics, epigenetics and lifestyle determinants [1,3,4]. Excessive fat accumulation in adipose tissue is related to a wide range of metabolic diseases, cardiovascular disorders, fatty liver and some types of cancer [5]. Therefore, more efficient therapeutic strategies targeting adipose tissue for prevention of obesity are needed [6].

Adipocytes are the main lipid storage cells in adipose tissue. White adipocytes store energy in the form of triglycerides and provide fatty acids and glycerol to other tissues, whereas brown adipocytes dissipate energy as heat in a process called thermogenesis [7], which is highly dependent on the activity of the uncoupling protein 1 (UCP1) [8]. Brown-like adipocytes have been described in the white adipose tissue (WAT), were named brite (browning-in-white) or beige, and respond to thermogenic stimuli [9]. Stimulating brown adipose tissue (BAT) activity and/or browning of WAT has been proposed as a strategy against overweight/obesity [10]. WAT expansion during obesity is due to an increase in the number (hyperplasia) and size (hypertrophy) of adipocytes [11]. In contrast, hyperplasia of BAT occurs mainly upon cold stimulation [12,13]. 

Hyperplasia and hypertrophy are mainly regulated by a family of proteins known as cyclin dependent kinases (CDKs), well-known for regulation of the cell cycle progression. CDKs are kinases whose activity depends on the formation of a complex with a regulatory subunit (cyclin) [14]. The CDK-cyclin complexes can be inhibited by CDK inhibitors (CDKIs) [15]. These CDKIs can be classified in two families: CIP/KIP inhibitors, which are able to inhibit a wide range of CDK-cyclin complexes, and the INK4 family, which specifically inhibit complexes formed by CDK4 or CDK6 [16]. Within the CIP/KIP family, CDKI 1B (CDKN1B), also known as *p27*^Kip1^, prevents the activation of the complexes formed by cyclin E-CDK2 and cyclin D-CDK4 by binding to them and stopping the cell cycle progression during the G1 phase [17,18]. A study showed that the overexpression of *p27* in Chinese hamster ovary cells (CHO) cells increased cell size, and was followed by an increase in oxygen and glucose consumption [19]. Moreover, a recent study of our group has shown that *p27* was overexpressed in the subcutaneous (sc)WAT of obese female mice and in the visceral WAT of obese men and women. In humans, higher expression of *p27* in scWAT was associated with a larger area of this depot [20]. This suggests that *p27* could have an important role in the regulation of the expansion of adipose tissue and its metabolism during obesity [20]. Interestingly, the *p27* knockout mouse model presents a markedly increased white adipose tissue mass [21], and in preadipocytes, progression of hyperplasia occurs upon a decrease of *p27* levels [22]. The absence of *p27* also caused a moderate increase of brown fat mass (25% vs. WT) [21]. In contrast, the overexpression of *p27* in adipose tissue led to an impairment of brown adipose activity tissue in mice, with a smaller BAT depot and less UCP1 [23]. Even though BAT accounts for a small proportion of whole-body fat mass, it has a crucial role in thermogenesis, body weight control and insulin sensitivity [24]. Therefore, it is necessary to expand in the mechanisms controlling BAT function. Additionally, several CDK complexes inhibited by *p27* have been related to glucose and adipose tissue metabolism [25,26,27]. Although the literature points to *p27* as a possible player in glucose and adipose tissue metabolism, its role is yet to be fully understood. In the present study we analysed the susceptibility of *p27*^−/−^ mice to develop HFD-induced obesity and insulin resistance. We also assessed the effects of whole-body *p27* deficiency on BAT morphology and function, and its potential relationship with changes in energy expenditure, respiratory exchange ratio and the development of obesity.

## 2. Results

### 2.1. p27^−/−^ Mice are More Prone to Develop Obesity Than WT Mice

Changes in body weight and body composition were assessed in WT and *p27* knockout mice fed with a normal chow diet (NCD) or a high fat diet (HFD) for 10 weeks. A two-way ANOVA test was used to establish if the obtained results were a consequence of the effect of the genotype (*p27* deficiency), of the diet, or of an interaction between both. Figure 1a shows that body weight gain was markedly enhanced by the lack of *p27* independently of the type of diet. Body composition analysis (Figure 1b) revealed that the HFD significantly increased the body fat mass and, interestingly, the lack of *p27* markedly stimulated body fat accumulation (weight and percentage), suggesting that *p27* deficiency confers susceptibility to develop obesity. Although *p27* null mice also exhibited a higher total lean mass content, the lean mass percentage was reduced, suggesting that the increase in fat mass is more important in contributing to the body weight gain in these animals. Indeed, the lack of *p27* significantly accounted for larger gonadal (gWAT), inguinal (iWAT) and interscapular brown (iBAT) adipose depots (Figure 1c). In addition, Figure 1d shows the weights of different organs, indicating that the lack of *p27* produced hepatomegaly, splenomegaly, cardiomegaly, nephromegaly and increased weight of the soleus muscle, but did not significantly affect the gastrocnemius muscle or the pancreas.

### 2.2. p27^−/−^ Mice Exhibit Increased Respiratory Exchange Ratio

To explore the possible causes of the increased susceptibility to diet-induced obesity of the *p27*^−/−^ mice, food intake (Appendix A), whole-body oxygen consumption (Figure 2a), energy expenditure (Figure 2b) and the respiratory exchange ratio (RER) (Figure 2c) were measured. In this regard, there were no significant differences on daily food intake between WT and *p27*^−/−^ mice, independently of feeding with NCD or HFD (Appendix A). Interestingly*, *p27**^−/−^ mice showed a tendency for reduced oxygen consumption and energy expenditure during the dark phase (Figure 2a,b). As expected, the animals fed with the HFD exhibited a reduced RER and, conversely, *p27* deficiency accounted for an increased total RER (Figure 2c), suggesting that *p27*^−/−^ mice may display a lower lipid oxidation capacity.

### 2.3. Mice Lacking p27 Are More Susceptible to HFD-Induced Insulin Resistance

It is well established that obesity plays a critical role in impairing glucose homeostasis and insulin resistance [28]. Therefore, we decided to analyse blood glucose and insulin levels. HFD in general elevated fasting serum glucose levels, which were not significantly affected by the lack of *p27* (Figure 3a). Concerning serum insulin levels, there was a significant interaction between the effects of the HFD and the lack of *p27* (Figure 3b). Indeed, *p27*^−/−^ NCD mice had elevated fasting serum insulin levels compared to the WT NCD mice, and WT HFD mice also tended to have increased insulin levels, suggesting the development of insulin resistance. Strikingly, *p27*^−/−^ HFD mice presented similar insulin levels to WT NCD mice (Figure 3b); however, the glucose levels were not normalized, suggesting that the reduced insulin levels in the *p27* null mice fed with a HFD are probably the consequence of deficient pancreatic insulin production as a result of a more aggravated insulin resistance. To further investigate this possibility, an insulin tolerance test (ITT) and a glucose tolerance test (GTT) were performed. In the ITT (Figure 3c), no differences were noticed between WT NCD and *p27*^−/−^ NCD groups. As expected, the WT HFD group exhibited impaired insulin response compared to WT NCD. Furthermore, *p27*^−/−^ HFD mice displayed severe insulin resistance when compared to the WT HFD mice (Figure 3c), suggesting that the lack of *p27* interferes with insulin signalling in the context of obesity. The differences were not as marked in the GTT (Figure 3d), which showed that the *p27*^−/−^ HFD mice tended to be more glucose intolerant than the WT HFD mice. However, there was no difference between the WT NCD and *p27*^−/−^ NCD regarding glucose tolerance. No influence of the *p27* deficiency was noticed on other serum parameters such as triglycerides, total cholesterol, LDL-cholesterol, HDL-cholesterol, β hydroxybutyrate, alanine aminotransferase (ALT) and aspartate aminotransferase (AST) (Table 1).

### 2.4. iBAT of p27^−/−^ HFD Mice Exhibits Whitening and an Impaired Insulin Signalling

BAT activation plays an important role in weight gain control because obesity is associated with decreased activity in BAT [29,30]. The BAT depot is also relevant for glucose homeostasis and insulin sensitivity [31]. Therefore, as the insulin resistance was exacerbated by the HFD in the mice lacking *p27* and the NCD had no effect, we focused on studying the effect of the absence of *p27* in combination with HFD. The interscapular BAT (iBAT) from the *p27*^−/−^ HFD mice was heavier (Figure 1c) and visually larger than their WT HFD counterparts (Figure 4a). Interestingly, the histological examination of the iBAT revealed a different morphology of the brown adipocytes between *p27*^−/−^ HFD and WT HFD mice, with larger and had bigger lipid droplets in the *p27* deficient mice, confirmed by the histological analysis of the brown adipocyte size (Figure 4b,c). This suggests that *p27* deficiency promotes brown adipocyte hypertrophy. However, considering the role of *p27* in cell cycle regulation, we cannot rule out that *p27* deficiency can also affect hyperplasia. Indeed, the mRNA expression of the *Ccna*, *Ccne* and *Cdk2*, which are involved in cell cycle progression and inhibited by *p27*, was significantly increased in *p27*^−/−^ HFD mice compared to WT HFD mice (Appendix A). Moreover, we tested the mRNA expression levels of two proliferation markers *Ki67* [13] and *Tpx2* [32], which were increased in the iBAT of *p27*^−/−^ HFD mice, suggesting that the increased size of iBAT observed in *p27*^−/−^ mice is secondary to both hyperplasia and hypertrophy (Appendix A). To obtain a broader image of the condition of the iBAT, the expression of genes relevant for different BAT functions was measured (Figure 4d). Interestingly, *Pparg* was augmented in *p27*^−/−^ HFD mice, suggesting increased adipogenesis. The lipogenesis-related gene, *Fas*, was downregulated and *Scd1* showed a similar tendency in *p27*^−/−^ HFD, probably due to the higher content of stored fat. Regarding fatty acid oxidation, *Acox* levels were downregulated, while no changes were observed in *Cpt1a* or in *Ppara*. There were no significant changes for genes typically involved in BAT development and activity (*Prdm16*, *Dio2*, *Pgc1a*). On the other hand, the glucose transporter *Glut1* was dramatically downregulated in *p27*^−/−^ HFD mice, but *Glut4* presented no change in these mice. However, a diminished expression of UCP1 at protein level was observed in iBAT of *p27*^−/−^ HFD mice (Figure 4e), which could lead to impaired thermogenic capacity. Although, the mRNA levels of the insulin-dependent glucose transporter *Glut4* were not affected by *p27* deficiency, it is important to note that the *p27*^−/−^ HFD mice exhibited impaired insulin response in iBAT, as demonstrated by the reduced phosphorylation of AKT, and therefore a reduced pAKT^S473^/AKT ratio (Figure 4f). All these data suggest that *p27*^−/−^ HFD mice may display impaired glucose uptake in BAT, which could also account for a defective thermogenic response. Similar to the results observed in iBAT, histological analysis of the iWAT revealed larger adipocytes in the *p27*^−/−^ HFD mice (Appendix A). Gene expression analysis revealed a marked downregulation of *Ucp1* and *Glut1* in this depot (Appendix A), which may also suggest a reduced capacity in the adaptive thermogenesis of this depot. Interestingly, the histological analysis of the liver and the determination of the hepatic triglyceride content, also suggest that *p27*^−/−^ HFD mice are more prone to accumulate fat and develop liver steatosis (Appendix A).

### 2.5. p27^−/−^ HFD Mice Have a Less Functional BAT and a Diminished Thermogenic Capacity

Since there was a decrease in UCP1 and *Glut1* as well as impaired insulin signaling observed in the BAT of *p27*^−/−^ HFD group, pointing towards a compromised thermogenic capacity in these mice, their response to cold exposure was explored. Upon the cold challenge, the *p27*^−/−^ HFD mice could not maintain their body temperature as did the WT HFD (Figure 5a), and after 2 h of cold exposure showed a higher decrease of body temperature (Figure 5b). Additionally, to better evaluate BAT functionality, a microPET study was carried out. ^18^F-fluorodeoxyglucose (^18^F-FDG) uptake was evaluated after one hour of cold exposure, showing a significant decrease in the ^18^F-FDG uptake of iBAT area in the *p27*^−/−^ HFD mice (Figure 5c), suggesting that the iBAT of the *p27*^−/−^ HFD mice had a lower capacity of taking up glucose than the WT HFD mice, which could be the reason for the defective response to adapt to a cold exposure challenge.

## 3. Discussion

This study supports that *p27* deficiency causes increased susceptibility to obesity, especially when mice are subjected to an obesogenic diet. This was demonstrated by the higher percentage of whole-body fat mass, and the larger size of all fat depots analysed (gWAT, iWAT and BAT). 

The increased size of the fat depots could be expected due, in part, to the absence of an appropriate regulation of the cell cycle progression, which would lead to higher cell proliferation rates, and greater growth and organomegaly, as previously described [33]. Indeed, it has been previously reported that loss of CDK inhibitors also produces adipocyte hyperplasia and obesity [21]. Furthermore, our current data suggest that the increase of adipose tissue mass could be due to the presence of hypertrophic adipocytes. Indeed, the analysis of iWAT and iBAT morphology clearly showed a higher adipocyte hypertrophy in the *p27* null mice fed with the HFD compared to the WT mice.

Interestingly, our study showed that *p27* deficient mice presented a trend for a moderate decrease of oxygen consumption and energy expenditure, which are typically observed in obesity [34]. Additionally, the increased RER levels observed in the *p27*^−/−^ mice suggest a reduced capacity for lipid oxidation, which could partly explain why fat accumulation in these mice is greater [35,36]. 

It is already known that BAT activity is critical for the improvement of insulin sensitivity and glucose metabolism [37]. In addition a lower BAT activity can contribute to bringing on obesity [38,39]. Therefore, a plausible explanation for the phenotype observed in the *p27*^−/−^ HFD mice could be defective BAT activity. First, we observed that *p27*^−/−^ HFD mice had a light brown coloured and larger and heavier iBAT depot with greater adipocyte size. The “whitening” of iBAT has already been described and causes a decline in its activity [40]. Expression analysis also revealed alterations in genes involved in peroxisomal fatty acid oxidation, such as *Acox*. Indeed, ACOX-dependent peroxisomal fatty acid oxidation can modulate BAT activity [41]. The downregulation of *Glut1* could also compromise the glucose uptake capacity in the iBAT of *p27*^−/−^ HFD mice, as it is highly dependent on GLUT1 transcription, synthesis and translocation to the membrane [42]. In fact, glucose is an important fuel source for BAT [43] and, as a result of the lowered uptake, BAT function is diminished. In addition, we explored insulin pathway activation, and the iBAT of the *p27*^−/−^ HFD mice exhibited a diminished pAKT^S473^/AKT ratio in response to acute insulin challenge, suggesting an impairment of the insulin signalling in this tissue, probably as a consequence of the higher fat accumulation [44]. Although the mRNA levels of the insulin-dependent glucose transporter *Glut4* were not altered in iBAT of the *p27* null mice, the alteration in insulin signalling suggests that the translocation of GLUT4 to the membrane to promote glucose uptake into the cell [45] could be also affected. Thus, both insulin-dependent and insulin-independent glucose transporters could be impaired in *p27* null mice, which may explain the reduced thermogenic capacity in these animals. 

Curiously, the relation between *p27* and glycaemia is rather contradictory. In this regard, a study [46] evaluated the effect on glucose metabolism of the deficiency of *p27*, finding that young *p27*^−/−^ mice fed with NCD had enhanced growth of pancreatic beta cells and elevated insulin levels, as occurred in the present study, together with improved glucose tolerance. In contrast, other studies have shown that 10- to 12-week-old *p27*^−/−^ mice were glucose intolerant and insulin insensitive [21], defining *p27* as a crucial mediator in beta cell generation [47]. Our study shows no changes in glucose or insulin tolerance in the *p27*^−/−^ mice fed on NCD, but the dramatic impairment of insulin response to the ITT observed in the *p27*^−/−^ HFD mice could be a consequence of the exhaustion of the pancreatic cells, as a significant reduction of the insulin circulating levels was observed in these mice [48] in combination with an increased resistance to insulin. 

Moreover, downregulation of UCP1 levels in the iBAT of the *p27*^−/−^ HFD mice, which is critical for BAT activity and body temperature maintenance [49], could compromise the cold adaptation capacity of *p27*^−/−^ HFD mice. In fact, these animals had a reduced capacity to keep their body temperature after cold exposure. This observation was combined with a lower iBAT glucose uptake capacity of the *p27*^−/−^ HFD in response to acute cold exposure. These facts suggest that the impairment of iBAT observed in *p27*^−/−^ mice could affect whole-body oxygen consumption [50], favouring increased body fatness and impaired insulin sensitivity and glucose metabolism. Indeed, a less active BAT is linked to a higher susceptibility to develop obesity and insulin resistance, and studies propose BAT activation as a strategy to fight these diseases [24]. However, studies assessing BAT activity and function in humans are complex, and cannot be carried out easily, therefore studies in mice have more relevance.

Our results suggest that the lack of *p27* impairs iBAT activity, which apparently contrasts with a previous study reporting that transgenic mice with adipose tissue-specific overexpression of *p27* showed a defect of body temperature maintenance when exposed to cold temperature. This circumstance was accompanied by a lower response of BAT to β-adrenergic stimulation [23]. The fact that the adipose-specific overexpression of *p27* results in a similar loss of BAT activity than the full-body ablation of *p27*, suggests that the impairment of BAT activity in these mice could be secondary to the alterations induced by the lack of *p27* in other tissues/systems with key roles in controlling the thermogenic activity of BAT, such as the sympathetic nervous system [51], which, if affected by the lack of *p27*, could alter the normal-functioning of this fat depot. In fact, it has already been reported that another cell cycle regulator, CDK4, modulates BAT thermogenesis via regulation of the sympathetic innervation of BAT through hypothalamic nuclei [52]. Furthermore, *p27* seems to be relevant in the hypothalamus [53] and for the development of the central nervous system [54] and its absence may disrupt its normal functioning. Thus, these observations suggest that cell cycle regulators might regulate BAT activity also acting at a nervous system level. However, further future studies are necessary to better clarify this. Indeed, the main limitation of this study is that a whole-body knockout mouse model was used. Therefore, it cannot be discerned whether the changes observed in iBAT activity and metabolic phenotype are caused directly by the lack of *p27* in adipose tissue, as effects in other non-adipose tissues are also probably involved. To clarify the role of *p27* in iBAT function and its relationship with whole-body metabolism, a study including brown adipose tissue-specific *p27* knockout mice should be carried out. It would be of interest to characterize this in parallel if the observations found at mRNA levels in *p27* null mice are also observed at protein or enzyme activity level.

Taken together, our data show that the whole-body deficiency of *p27* favors the development of obesity, increasing the size of all fat depots independently of the diet. The deficiency of *p27* combined with a HFD causes severe insulin resistance. In iBAT, *p27* deficiency, together with HFD, promotes adipocyte hypertrophy, while it decreases the mRNA levels of the glucose transporter *glut1* and disrupts the insulin signaling pathway (pAKT/AKT ratio). Moreover, *p27*^−/−^ HFD mice display lower levels of the thermogenic protein UCP1 in iBAT and an impaired response to cold exposure in parallel with a difficulty to maintain body temperature, suggesting reduced thermogenic capacity. These data suggest that the cell cycle inhibitor *p27* could play a role in adipose tissue metabolism, especially in BAT. This highlights the importance of investigating the metabolic functions of *p27*, and other cell cycle regulators, with respect to their function in BAT and the development of obesity. 

## 4. Materials and Methods

### 4.1. Animals

*The genetic background for WT and *p27*^−/−^ mice was C57/BL6J.* The *p27*^−/−^ mice were provided by Dr. Philip Kaldis [55,56]. WT and *p27*^−/−^ male and female mice were obtained by crossing mice heterozygous for *p27* (*p27*^+/−^). After weaning, mice were housed in cages in groups of 4–8 animals under controlled conditions (22 ± 2 °C, with a 12 h light-dark cycle, relative humidity, 55 ± 10%). At 6–8 weeks of age, animals were fed either with a standard pelleted normal chow diet (NCD, 13% of kcal from fat, 67% from carbohydrates and 20% from proteins, Harlan Teklad Global Diets, Harlan Laboratories, Indianapolis, IN, USA) or a HFD (45% kcal from fat, 35% form carbohydrates and 20% from proteins, Research Diets, New Brunswick, NJ, USA) for 10 weeks.

At the endpoint, body composition was measured by magnetic resonance technology (EchoMRI-100-700; Echo Medical Systems, Houston, TX, USA), as previously described [57]. Mice were sacrificed, blood was extracted, and serum samples were obtained. Different tissues and organs (gWAT, iWAT, iBAT, liver, spleen, gastrocnemius, heart, pancreas, and kidney) were immediately collected, weighed and snap-frozen in liquid nitrogen and kept at −80 °C for further determinations, or fixed with 4% formalin to perform histological analyses. Experiments and analyses were performed in both male and female mice, except for the in vivo study of iBAT activation by positron emission tomography (PET), which was carried out only in males All experimental procedures were performed under protocols approved by the University Ethics Committee for the use of laboratory animals according to the national and institutional guidelines for animal care and use (protocols: 048-17, E41-19 (048-17E2), 073-22, 049-17, 072c-22).

### 4.2. Insulin and Glucose Tolerance Test

For the ITT, a week prior to sacrifice mice were fasted for 6 h, and baseline blood glucose levels were measured with a standard glucometer (Accu-Check Advantage blood glucose meter, Roche, Manheim, Germany). Animals were then injected (i.p. 0.375 mU/g of body weight) with human rapid insulin (Actrapid^®^ Innolet^®^, Novo Nordisk A/S, Bagsvaerd, Denmark) and blood glucose levels were measured 30, 60 and 120 min after injection. For the GTT, mice were fasted overnight and after measuring fasted blood glucose levels, mice were injected i.p. with 1 g glucose/kg of body weight (Merck KGaA, Darmstadt, Germany) and blood glucose levels were then measured 30, 60 and 120 min after the injection.

### 4.3. Biochemical Analysis

Fasted glucose serum levels were measured with a standard glucometer (Accu-Check Advantage blood glucose meter, Roche) and serum levels of total cholesterol, HDL-cholesterol, triglyceride, β hydroxybutyrate, AST and ALT were determined using a Pentra C200 autoanalyser (Roche Diagnostic, Basel, Switzerland) following the manufacturer’s instructions, as described elsewhere [58]. LDL-cholesterol was calculated using Friedewald’s equation [59]. Serum insulin was measured using a commercially available ELISA kit (Mercodia, Uppsala, Sweden), according to the manufacturer’s guidelines.

### 4.4. Insulin Signalling Assessment

To evaluate the activation of the insulin signalling pathway, overnight fasted mice were injected i.p. with human rapid insulin (Actrapid^®^ Innolet^®^, Novo Nordisk A/S) (1 mU/g body weight) and sacrificed 15 min later. Then, fat depots were excised and frozen in liquid nitrogen. The insulin signalling pathway was evaluated by measuring the phosphorylation of AKT^S473^ vs. total AKT levels by Western blot (see below).

### 4.5. Real-Time Quantitative PCR

Total RNA from mouse samples was extracted with TRIzol™ reagent (Invitrogen, ThermoFisher Scientific, Waltham, MA, USA) according to the manufacturer’s instructions. RNA quality and concentrations were measured by Nanodrop Spectrophotometer ND1000 (Nanodrop Technologies, Inc., Wilmington, NC, USA). RNA (1–5 µg) was then incubated with DNase I (RapidOut DNA Removal kit, Thermo Fisher Scientific) for 30 min at 37 °C and reverse transcribed to cDNA using the High-Capacity cDNA Reverse Transcription Kit (Applied Biosystems; Waltham, MA, USA, Thermo Fisher Scientific) according to the manufacturer’s instructions in a Touch PCR system (C1000, BIO-RAD, Hercules, CA, USA). Real-time PCR was performed using the Touch Real-Time PCR System (CFX384, BIO-RAD). The expression of genes was determined using Power SYBR Green PCR Master Mix (BIO-RAD) or TaqMan master mix (Applied Biosystems). SYBR Green primers were tested with Primer-Blast software (National Center for Biotechnology Information, Bethesda, MD, USA; https://www.ncbi.nlm.nih.gov/tools/primer-blast, accessed on 13 January 2023). Primer sequences are shown in Table 2. For relative quantitation of gene expression, the comparative Ct method was used [2^−ΔΔCt^, where ΔCt represents the difference in threshold cycle between the target and a housekeeping gene (*36b4*)] [60].

### 4.6. Protein Extraction and Western Blot

Tissues were thawed and homogenized in lysis buffer [RIPA buffer Pierce™ (Thermo Scientific, 25 mM Tris·HCl pH 7.6, 150 mM NaCl, 1% NP-40, 1% sodium deoxycholate, 0.1% SDS and protease inhibitors cocktail (Roche)], centrifuged, and protein concentrations were determined by Pierce™ BCA Protein Assay Kit (Thermo Fischer Scientific). Proteins were separated with 12% SDS-PAGE, transferred to nitrocellulose membranes, and stained with Ponceau-S red solution (Thermo Fischer Scientific) to verify equal loading of proteins. The membranes were probed with primary antibodies against phospho-AKT (S473, rabbit Cell Signalling Technologies, Danvers, MA, USA), AKT (rabbit, Cell Signalling Technologies), UCP1 (rabbit, Abcam, Cambridge, UK) and β-actin (mouse, Sigma-Aldrich, St. Louis, MO, USA). Briefly, membranes were blocked for 2 h in TBS-Tween-20 [50 mM Tris-HCL (pH 7.6), 200 mM NaCl and 0.1% Tween-20] with 5 % BSA (Sigma-Aldrich), and then primary antibody (1:1000) was added and incubated overnight. Thereafter, membranes were rinsed and incubated with the secondary antibody for 1 h at room temperature. For UCP1 and β-actin infrared fluorescent secondary antibody anti-rabbit (1–10,000) or anti-mouse (1–10,000) (Cell Signalling Technology) were used and quantitated using an Odyssey Sa infrared imaging system (Imagen Studio Lite; LI-COR Biosciences, Lincoln, NE, USA). For AKT and pAKT^S473^ the immunoreactive proteins were detected with enhanced chemiluminescence with the corresponding peroxidase conjugated secondary antibody (Cell Signalling Technologies) at 1:10,000 using the C-DiGit^®^ Blot Scanner (LI-COR) and quantified by densitometry analysis (Imagen Studio Lite; LI-COR Biosciences). 

### 4.7. Acute Cold Exposure Challenge

Mice were housed at 4 °C for 2 h, and rectal temperature was measured at basal state and every 30 min upon cold exposure rectal thermometer probe (Panlab, Barcelona, Spain) [61].

### 4.8. In Vivo Study of iBAT Activation by Positron Emission Tomography (PET)

iBAT activation was studied in vivo by microPET with the radiotracer ^18^F-FDG. BAT was stimulated by cold exposure for 1 h at 4 °C [62]. After cold exposure, ^18^F-FDG (10.1 ± 0.9 MBq) was injected through the tail vein. MicroPET static images were acquired at room temperature 1 h after^18^F-FDG injection in a small animal microPET scanner (Mosaic, Philips, Amsterdam, The Netherlands). For each study, animals were anesthetized with 2% isoflurane in 100% O_2_ gas. All studies were analysed using PMOD software (PMOD Technologies Ltd., Adliswil, Switzerland). For semiquantitative analysis, ^18^F-FDG uptake by iBAT was evaluated drawing volume-of-interest (VOIs) on coronal PET images. From each VOI, the maximum standardized uptake value (SUVmax) was calculated using the formula SUV = (tissue activity concentration (Bq cm^3^)/injected dose (Bq)] × body weight (g)).

### 4.9. Indirect Calorimetry and Activity Measurements

Animals were placed in a comprehensive laboratory animal monitoring system for measurements at 20 °C attached to custom-built oxygen and carbon dioxide monitoring system (PhenoMaster, TSE Systems, Berlin, Germany). Airflow rates were 400 mL/min for measurements of oxygen concentration, and carbon dioxide concentrations in room air and air leaving each the cage were measured every 20 min as described before [63]. Energy expenditure (EE) was calculated from the amount of oxygen consumed (VO_2_) and the amount of carbon dioxide produced (VCO_2_) using the equation: EE (J) = 15.818 VO_2_ + 5.176 VCO_2_. The RER was calculated as the ratio of VCO_2_/VO_2_. Food intake of each animal was also evaluated using these metabolic cages. Indirect calorimetry studies were carried out in the LAFEMEX Laboratory (Universidad Rey Juan Carlos, Alcorcón, Spain).

### 4.10. Histological Analysis

Sections of paraffin-embedded tissues were stained with H&E. Representative images were taken using a Nikon eclipse e800 microscope and a Nikon DS-Qi1Mc camera at 20× and 40× (Nikon, Tokyo, Japan). Adipocyte cell size was determined using ImageJ software [64]. 

### 4.11. Liver Triglyceride Content Assay

Liver pieces (50 mg) were homogenized in buffer (Tris 10 mM, EDTA 2 mM, Sacarose 0.25 M; pH = 7) to determine TG content of the samples. Following the manufacturer’s instructions, the InfinityTM Triglycerides Liquid Stable Reagent (Thermo Fisher Scientific) was used to hydrolyze the TGs of the samples. This method is based on that of Wako with the modifications of McGowan et al. [65] and Fossati et al. [66]. The triglyceride concentration was normalized to protein content.

### 4.12. Statistical Analysis

Statistical analyses were carried out with GraphPad Prism 9 software (Graph-Pad Software, La Jolla, CA, USA). Data are presented as mean ± SEM. The effects of “Genotype” (*p27* deficiency) and “Diet” were analysed by two-way ANOVA. If an interaction was found between the two factors (Genotype × Diet), Student’s *t*-test was used for further analysis of differences between groups. Comparisons between two groups were analysed by Student’s *t*-tests or Mann Whitney’s tests according to parametric or nonparametric distribution, respectively. Differences between groups were set up as statistically significant at *p* < 0.05.

## Figures and Tables

**Figure 1 ijms-24-02664-f001:**
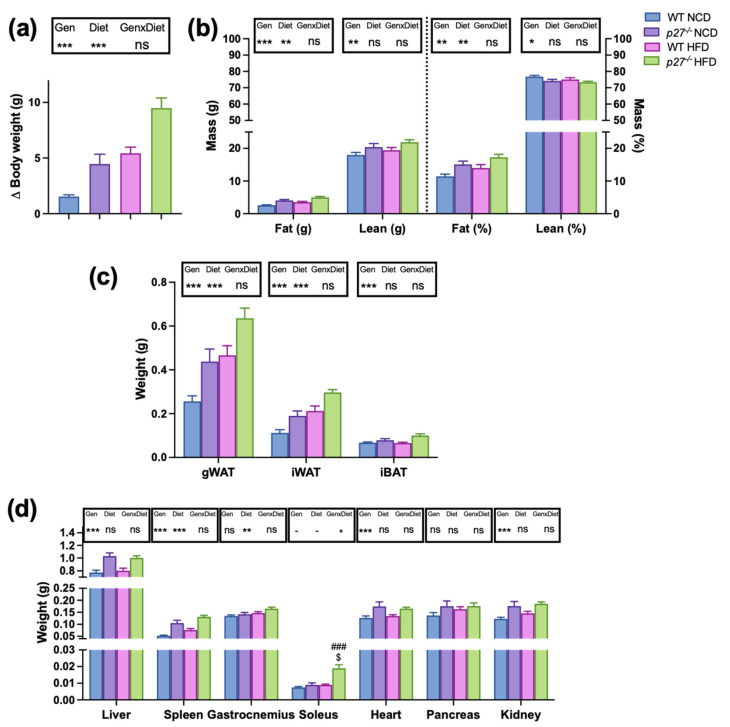
Mice lacking *p27* gain more weight and have larger WAT and BAT depots than WT mice. (**a**) Body weight gain after 10 weeks of study in WT and *p27*^−/−^ mice fed with normal chow diet (NCD) or a high fat diet (HFD). (**b**) Whole body fat and lean mass (in g or as percentage of body weight) in WT and *p27*^−/−^ mice fed with NCD or HFD for 10 weeks. (**c**) Weights of different adipose tissue depots: gonadal (gWAT) and inguinal (iWAT) and interscapular brown (iBAT) collected after the sacrifice in WT and *p27*^−/−^ mice fed with NCD or HFD for 10 weeks. (**d**) Weights of different organs collected after the sacrifice in WT and *p27*^−/−^ mice fed with NCD or HFD for 10 weeks. Data are expressed as mean ± SEM (*n* = 11–25). The legend above subfigures shows the result of the two-way ANOVA analysis (* *p* < 0.05, ** *p* < 0.01, *** *p* < 0.001, ns: non-significant). Gen, genotype; GenxDiet, interaction between genotype and diet. If an interaction (GenxDiet) was found, differences between groups were analysed by Student’s *t*-test. ^###^ *p* < 0.001 vs. WT HFD. ^$^ *p* < 0.05 vs. *p27*^−/−^ NCD.

**Figure 2 ijms-24-02664-f002:**
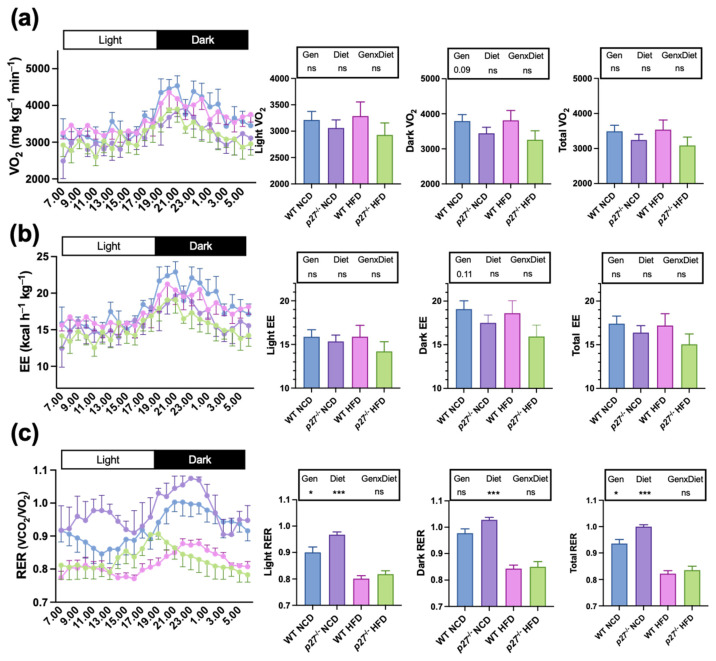
Mice lacking *p27* have increased respiratory exchange ratio (RER). (**a**) Oxygen consumption (VO_2_). (**b**) Energy expenditure. (**c**) Respiratory exchange ratio (RER) measured by indirect calorimetry in WT and *p27*^−/−^ mice fed with NCD or HFD for 10 weeks. Data are expressed as mean ± SEM (*n* = 4–7). Legend above subfigures shows the result of the two-way ANOVA analysis; ** p* < 0.05*, *** p* < 0.001, ns: non-significant. Gen, genotype; GenxDiet, interaction between genotype and diet.

**Figure 3 ijms-24-02664-f003:**
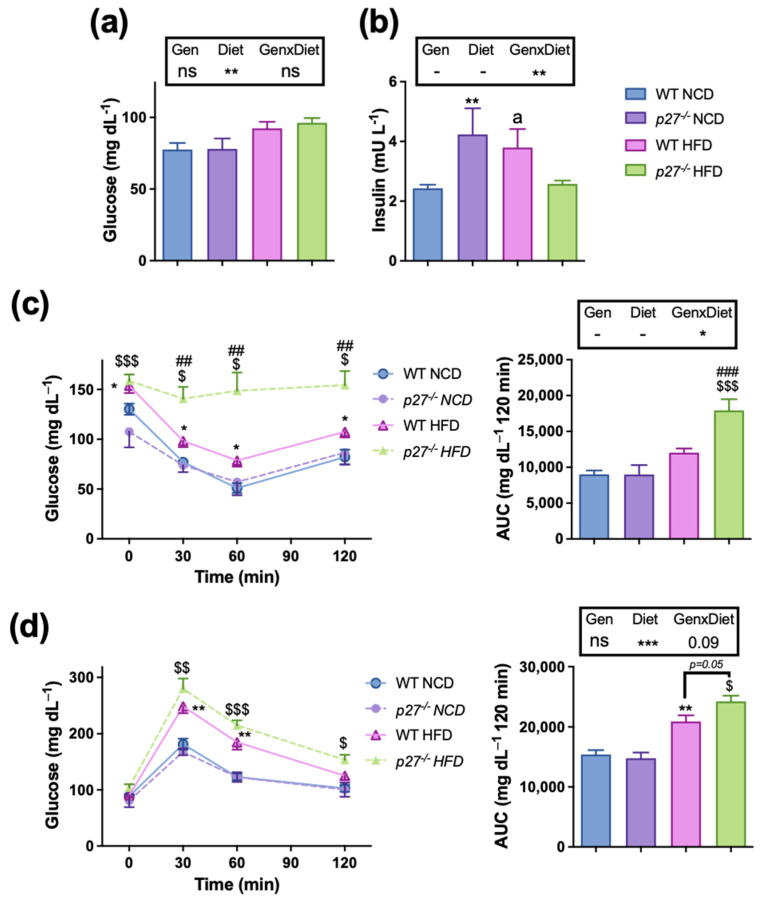
*p27*^−/−^ mice on HFD exhibit severe insulin resistance. (**a**,**b**) Fasting serum glucose (**a**) and insulin (**b**) in WT and *p27*^−/−^ mice fed with NCD or HFD for 10 weeks. (**c**) An insulin tolerance test (ITT) was performed after 6 h of fasting, and then mice were injected i.p. with 0.375 mU/g of body weight of rapid insulin and blood glucose levels were measured before and 30, 60 and 120 min after injection. (**d**) A glucose tolerance test performed after overnight fasting, and then mice were injected i.p. with 1 g/kg of body weight of glucose and blood glucose levels were measured before and 30, 60 and 120 min after injection. Data are expressed as mean ± SEM (*n* = 5–27). The legend above subfigures shows the result of the two-way ANOVA analysis; * *p* < 0.05, ** *p* < 0.01, *** *p* < 0.001, ns: non-significant. If an interaction (GenxDiet) was found, differences between groups were analysed by Student’s *t*–test. * *p* < 0.05, ** *p* < 0.01 vs. WT NCD; ^##^ *p* < 0.01, ^###^ *p* < 0.001, ^a^ *p* = 0.05 vs. WT HFD. ^$^ *p* < 0.05, ^$$^ *p* < 0.01, ^$$$^ *p* < 0.001 vs. *p27*^−/−^ NCD. Gen, genotype; AUC, Area Under the Curve.

**Figure 4 ijms-24-02664-f004:**
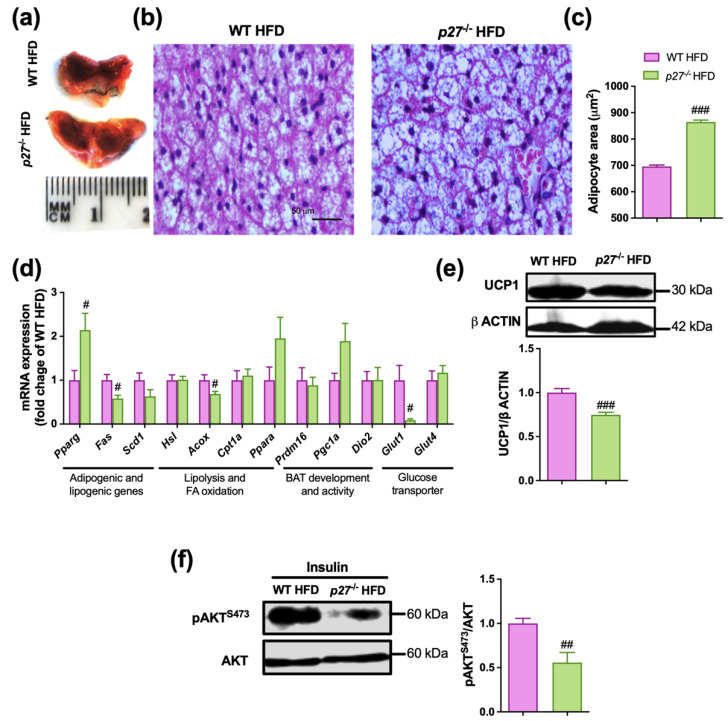
iBAT of *p27*^−/−^ HFD mice exhibits whitening and impaired insulin signalling. (**a**–**c**) Representative macroscopic (**a**) and histological images (**b**) and adipocyte area quantification (**c**) of iBAT in WT and *p27*^−/−^ mice fed with HFD for 10 weeks. (**d**) mRNA expression analysis of genes with relevant roles in iBAT function/metabolism in WT and *p27*^−/−^ mice fed with HFD for 10 weeks. (**e**) Representative image and densitometric quantification of Western blot analysis of UCP1 in iBAT of WT and *p27*^−/−^ mice fed with HFD for 10 weeks. (**f**) Representative image and densitometric quantification of Western blot analysis of pAKT^S473^/AKT in iBAT of insulin-stimulated WT and *p27*^−/−^ mice fed with HFD for 10 weeks. Data are expressed as mean ± SEM (*n* = 5–9). ^#^ *p* < 0.05, ^##^ *p* < 0.01, ^###^ *p* < 0.001 vs. WT HFD.

**Figure 5 ijms-24-02664-f005:**
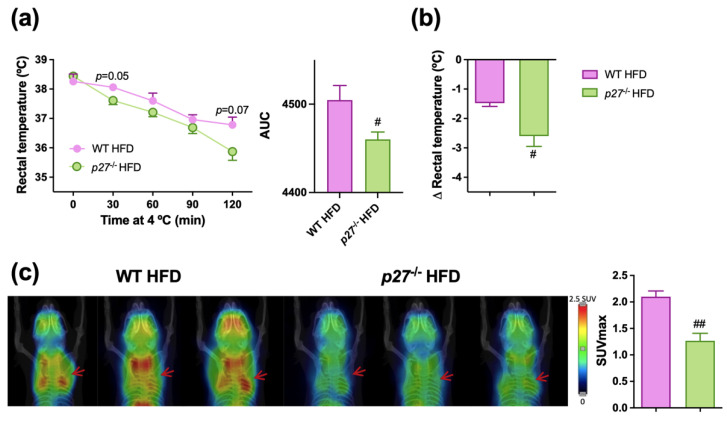
*p27*^−/−^ HFD mice exhibit lower iBAT activity and impaired response to cold exposure. (**a**,**b**) Rectal temperature measured at different timepoints during cold exposure to 4 °C including AUC (area under the curve) (**a**) and changes in rectal temperature before and after 2 h of cold exposure (**b**) in WT and *p27*^−/−^ mice fed with HFD for 10 weeks (*n* = 5–10). (**c**) ^18^F-FDG uptake in iBAT using microPET 1 h after ^18^F-FDG injection in WT and *p27*^−/−^ male mice fed with HFD for 10 weeks and pre-exposed for 1 h at 4 °C. Left panel: representative microPET coronal slices of the iBAT area (red arrow) co-registered with a CT 3D image of another animal used as anatomical reference. Right panel: maximum standardized uptake value (SUVmax) (*n* = 3). Data are expressed as mean ± SEM. ^#^ *p* < 0.05, ^##^ *p* < 0.01, vs. WT HFD.

**Table 1 ijms-24-02664-t001:** Serum biochemical parameters in WT and *p27*^−/−^ mice fed with NCD or HFD for 10 weeks.

	WT NCD	*p27*^−/−^ NCD	WT HFD	*p27*^−/−^ HFD	Gen	Diet	GenxDiet
Total cholesterol (mg/dL)	112.40 ± 5.61	102.80 ± 7.77	122.00 ± 7.20	121.20 ± 7.11	ns	ns	ns
LDL-cholesterol (mg/dL)	42.90 ± 3.13	43.34 ± 7.63	47.34 ± 3.57	47.57 ± 4.22	ns	ns	ns
HDL-cholesterol (mg/dL)	57.27 ± 3.00	43.94 ± 2.23	56.45 ± 3.86	54.54 ± 3.61	ns	0.07	ns
β Hydroxybutyrate (mg/dL)	1.92 ± 0.27	1.67 ± 0.47	1.38 ± 0.31	0.83 ± 0.13	ns	*	ns
Triglycerides (mg/dL)	91.25 ± 7.30	77.60 ± 8.37	95.56 ± 6.83	93.56 ± 5.61	ns	ns	ns
ALT (U/L)	41.36 ± 2.75	40.44 ± 1.36	46.50 ± 3.50	55.28 ± 4.91	ns	*	ns
AST (U/L)	278.20 ± 30.16	240.00 ± 20.12	397.90 ± 45.81	478.10 ± 49.82	ns	*	ns

Data are expressed as mean ± SEM (*n* = 5–18). Results of the two-way ANOVA analysis are represented in the table; * *p* < 0.05, ns: non-significant. Gen, genotype; GenxDiet, interaction between genotype and diet. ALT, Alanine aminotransferase; AST, Aspartate aminotransferase; LDL-cholesterol, low density lipoprotein cholesterol; HDL-cholesterol, high density lipoprotein cholesterol.

**Table 2 ijms-24-02664-t002:** Sequences of primers used in this study.

Gene	Species	Sequence/Reference
*Pparg*	*Mus musculus*	Mm00440940_m1
*Fas*	*Mus musculus*	Fw: GCTGCGGAAACTTCAGGAAAT
		Rv: AGAGACGTGTCACTCCTGGACTT
*Scd1*	*Mus musculus*	Mm00772290_m1
*Hsl*	*Mus musculus*	Fw: CTGCTTCTCCCTCTCGTCTG
		Rv: CAAAATGGTCCTCTGCCTCT
*Acox*	*Mus musculus*	Fw: CTATGGGATCAGCCAGAAAG
		Rv: AGTCAAAGGCATCCACCAA
*Cpt1a*	*Mus musculus*	Fw: CACCAACGGGCTCATCTTCTA
		Rv: CAAAATGACCTAGCCTTCTATCGAA
*Ppara*	*Mus musculus*	Rv: TCAGGGTACCACTACGGAGT
		Fw: CTTGGCATTCTTCCAAAGCG
*Prdm16*	*Mus musculus*	Rv: CAGCACGGTGAAGCCATTC
		Fw: GCGTGCATCCGCTTGTG
*Pgc1a*	*Mus musculus*	Fw: CTAGCCATGGATGGCCTATTT
		Rv: GTCTCGACACGGAGAGTTAAAG
*Dio2*	*Mus musculus*	Mm00515664_m1
*Glut1*	*Mus musculus*	Fw: TCAACACGGCCTTCACTG
		Rv: CACGATGCTCAGATAGGACATC
*Glut4*	*Mus musculus*	Fw: AAAAGTGCCTGAAACCAGAG
		Rv: TCACCTCCTGCTCTAAAAGG
*36b4*	*Mus musculus*	Fw: CACTGGTCTAGGACCCGAGAAG
		Rv: GGTGCCTCTGGAGATTTTCG
*Cdk2*	*Mus musculus*	Fw: GGGTCCATCAAGCTGGCAGA
		Rv: CCACAGGGTCACCACCTCAT
*Ccne*	*Mus musculus*	Fw: GCATCAGTATGAGATTAGGAATTG
		Rv: CAGAATGCAGAACTTGAAAATGT
*Ccna*	*Mus musculus*	Rv: CAAGACTCGACGGGTTGCTC
		Fw: GCTGGCCTCTTCTGAGTCTC
*MKi67*	*Mus musculus*	Rv: AATCCAACTCAAGTAAACGGGG
		Fw: TTGGCTTGCTTCCATCCTCA
*Tpx2*	*Mus musculus*	Fw: CCGAGTGCCCATCAAAGATC
		Rv: ATGGTGTCAAAATGGGGCAC

*Pparg*: Peroxisome Proliferator Activated Receptor Gamma; *Fas*: Fatty Acid Synthase; *Scd1*: Stearoyl-CoA desaturase-1; *Hsl*: Hormone Sensitive Lipase; *Acox*: Acyl-CoA Oxidase 1; *Cpt1a*: Carnitine Palmitoyltransferase 1A; *Ppara*: Peroxisome Proliferator Activated Receptor Alfa; *Prdm16*: PR/SET Domain 16; *Pgc1a*: Peroxisome proliferator-activated receptor gamma coactivator-1 alpha; *Dio2*: Iodothyronine Deiodinase 2; *Glut1*: Glucose Transporter 1; *Glut4*: Glucose Transporter 4; *Cdk2*: Cyclin-dependent kinase 2; *Ccne*: cyclin e; *Ccna:* cyclin a; *MKi67*: Marker Of Proliferation Ki-67; *Tpx2*: TPX2, microtubule-associated.

## Data Availability

Not applicable.

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
