# Peer review of "p27Kip1 Deficiency Impairs Brown Adipose Tissue Function Favouring Fat Accumulation in Mice"

_ijms, 2023, doi:10.3390/ijms24032664_

Round 1
Reviewer 1 Report
Colón-Mesa et al., explored the role of p27 on brown adipose tissue under normal chow diet and high fat diet. There are some merits. But some questions need to be clarified before acceptance.
Obesity is mainly caused by the increase of white adipose tissue. Brown adipose tissue accounts for a small amount of body mass. Besides, p27-/- mice gained fat mass mainly due to white adipose tissue why do you choose brown adipose tissue as the target? Please clarify in the introduction and in the discussion.
Lean mass is increased in p27-/- mice, but no difference is observed in gastrocnemius muscle. What might be the reason? Did you measure the muscle mass such as quadricep? Muscle is also major tissue for energy metabolism, can you rule out that the changes of overall energy metabolism is mainly due to the changes of adipose tissue? Same question for GTT and ITT. Muscle, liver and adipose tissue are insulin sensitivity organs. Did you check if liver morphology is affect by p27? Because you used a whole body knockout mouse model, please mention the limitations of your study such as you can't really rule out the overall changes is only caused by the adipose tissue.
The sex of the mice used in this study need to be clarified.
Line 71-74 references need to be included.
Blot proteins related lipid metabolism will be helpful to make a solid conclusion.
Reviewer 2 Report
The manuscript submitted for review is innovative and up to date. The study design is appropriate and results are clearly presented. However, there are some drawbacks needed to be addressed:
1. The aim of the study presented in the Abstract is not clear. It describes what was done, but not why it was done. Similarly, the purpose in the Introduction should put more attention to significance of the obtained results.
2. The major findings of the study revealed that knockout of p27 in mice evoked the development of obesity, insulin resistance and caused greater fat depots. The disturbances in "browning" of white adipcytes or "whitening" of brown adipocytes were suggested. Could you explain whether these findings maybe connected with the fact that p27 plays significant role in cell cycle and proliferation. Please provide the molecular mechanisms.
3. The authors did not propose a clinical significance of their findings (impaired BAT activation).
4.What further investigations should be performed to verify the hypothesis put in the second to last paragraph.
5. Conclusions are scrace.
